# Diester Chlorogenoborate Complex: A New Naturally Occurring Boron-Containing Compound

**Andrei Biţă** [1,2], **Ion Romulus Scorei** [1,*], **Nagendra Rangavajla** [3], **Ludovic Everard Bejenaru** [2], **Gabriela Rău** [1,4], **Cornelia Bejenaru** [5], **Maria Viorica Ciocîlteu** [1,6], **Laura Dincă** [1,7], **Johny Neamţu** [1,8], **Andrei Bunaciu** [9], **Otilia Constantina Rogoveanu** [10], **Mihai Ioan Pop** [11] and **George Dan Mogoşanu** [1,2]

1. Department of Biochemistry, BioBoron Research Institute, S.C. Natural Research S.R.L., 31B Dunării Street, Dolj County, 207465 Podari, Romania; andreibita@gmail.com (A.B.); gabriela.rau@umfcv.ro (G.R.); mariaviorica.bubulica@gmail.com (M.V.C.); eu-office@umfcv.ro (L.D.); johny.neamtu@umfcv.ro (J.N.); george.mogosanu@umfcv.ro (G.D.M.)
2. Department of Pharmacognosy & Phytotherapy, Faculty of Pharmacy, University of Medicine and Pharmacy of Craiova, 2 Petru Rareş Street, Dolj County, 200349 Craiova, Romania; ludovic.bejenaru@umfcv.ro
3. Califia Farms, 1321 Palmetto Street, Los Angeles, CA 90013, USA; rangavan@gmail.com
4. Department of Organic Chemistry, Faculty of Pharmacy, University of Medicine and Pharmacy of Craiova, 2 Petru Rareş Street, Dolj County, 200349 Craiova, Romania
5. Department of Pharmaceutical Botany, Faculty of Pharmacy, University of Medicine and Pharmacy of Craiova, 2 Petru Rareş Street, Dolj County, 200349 Craiova, Romania; cornelia.bejenaru@umfcv.ro
6. Department of Analytical Chemistry, Faculty of Pharmacy, University of Medicine and Pharmacy of Craiova, 2 Petru Rareş Street, Dolj County, 200349 Craiova, Romania
7. Department of International Relations, University of Medicine and Pharmacy of Craiova, 2 Petru Rareş Street, Dolj County, 200349 Craiova, Romania
8. Department of Physics, Faculty of Pharmacy, University of Medicine and Pharmacy of Craiova, 2 Petru Rareş Street, Dolj County, 200349 Craiova, Romania
9. S.C. AAB_IR Research S.R.L., 9–11A Gloriei Street, Ilfov County, 077025 Bragadiru, Romania; aabunaciu@gmail.com
10. Department of Physical Medicine and Rehabilitation, University of Medicine and Pharmacy of Craiova, 2 Petru Rareş Street, Dolj County, 200349 Craiova, Romania; otilia.rogoveanu@umfcv.ro
11. Faculty of Medical Sciences, University College London, Gower Street, Bloomsbury, London WC1E 6AB, UK; popmihaiioanbm@gmail.com
* Correspondence: romulus_ion@yahoo.com; Tel.: +40-351-407-543

**Abstract:** The natural compounds of boron have many applications, primarily as a dietary supplement. The research is based on the discovery that the diester chlorogenoborate complex can be detected and quantified from green coffee beans. The study reports that such a diester molecule can also be synthesized in a stable form via the direct reaction of boric acid and chlorogenic acid in a mixture of acetonitrile–water (1:1, v/v) and left to evaporate over a period of 48 h at room temperature, resulting in a spirocyclic form (diester complex). The diester complex, with its molecular structure and digestibility attributes, has potential application as a prebiotic in gut health and oral health, and as a micronutrient essential for microbiota in humans and animals.

**Keywords:** diester chlorogenoborate complex; green coffee bean; HPTLC/UV; HPTLC/ESI–MS; UHPLC/MS; FTIR

## 1. Introduction

Boron (B) organic species are present in plants across a large range of essential primary metabolites including B–carbohydrate complexes [1,2] and B amino acids [3,4], as well as secondary metabolites, such as organic acids [5,6], and are still undiscovered in vivo as B phenolic compounds [7,8].

Phenolic acids are found in plants and exert a significant biological function in achieving communication between plants and other organisms [9,10]. At the same time, phenolic

acids act on microbiota through the suppression of predators and pathogens, as well as the stimulation of mycorrhizae [11–13].

Chlorogenic acids (CGAs—according to the International Union of Pure and Applied Chemistry (IUPAC) nomenclature) are esters of caffeic and quinic acids, and there are numerous types of polyphenols found in the nutrition of humans. CGA isomers found in the green coffee bean (GCB) consist of the following: 3-caffeoylquinic acid (3-CQA), 4-caffeoylquinic acid (4-CQA), and 5-caffeoylquinic acid (5-CQA). The 5-CQA is the most abundant CGA identified in GCB, with a percentage of 76–84% of the total CGAs. 5-CQA (IUPAC nomenclature) is a phenolic compound that has demonstrated numerous activities effective in human health, such as antioxidant, anti-inflammatory, antilipidemic, antidiabetic, antihypertensive, and antimicrobial actions [14]. Its metabolism by colonic microflora has also been documented, and it appears to be a prebiotic for commensal bacteria including Bifidobacterium spp. and it is also thought to be responsible for some of the health-promoting effects of coffee [15].

CGAs have shown activity as a prebiotic. Coffee with high levels of CGA (high-CGA coffee) determines an important increase in the development of the Bifidobacterium spp. and the Clostridium coccoides–Eubacterium rectale group [16]. CGAs alleviate the damage of the colon mucus produced by a rich-fat diet via gut microflora regulation to boost short-chain fatty acids (SCFAs) growth in rats. CGAs ameliorate colitis and alter the microbiota of the colon in a model of dextran sulfate sodium-induced colitis in mice. CGAs can regulate the abundance and diversity of the gut microbial population [17].

The high B complexation capacity of CQAs and the use of CQAs in the analytical chemistry of B determination [18,19] have led us to search in plants with high concentrations of B and CQAs, as it is well known that phenol metabolism needs B for healthy plant metabolism. It is also known that the lack of B in the metabolism of phenols in the plant causes their oxidation. At the same time, the formation of B complexes determines phenolics' unavailability to oxidation [20–22].

In the literature, it has been established that (*i*) the coupling of phenolic acids (3,4-dihydroxybenzoic acid, caffeic acid, and CQAs) and boric acid (BA), when observed by spectroscopy, yields a 1:1 assembly in every reaction [7]; (*ii*) with the first two acids (3,4-dihydroxybenzoic acid and caffeic acid), there is proof that a 2:1 compound also resulted in a significant acid-to-B proportion; however, for CQAs, no 2:1 diester complexes were identified [7]; (*iii*) catechol derivatives and caffeic acid only form mono-borate compounds [23]. Subsequently, existing studies do not signal the in vitro formation of chlorogenic diester with B. Previous work did not find evidence for the formation of 1:2 BA complexes with catechol (B phenolic diesters) and substituted catechol that had been examined.

Our communication is based on the identification and semisynthesis of a new natural B phenolic compound—diester chlorogenoborate (5-DCB) complex (IUPAC nomenclature)—which, being a complex of B with a phenolic acid, can be nutritive prebiotic for human and animal microbiota [24]. We chose only plants with a B concentration higher than 10 ppm and high phenolic acids to increase the possibility of identifying new B compounds (especially B-containing phenolic acid compounds). After the rigorous tests that were highlighted in a previous report, we chose only GCB as it contained the highest amount of B organic species and CQAs [25].

We present here a method for 5-DCB semisynthesis and rapid identification from GCB by using the high-performance thin-layer chromatography (HPTLC)/ultraviolet (UV) densitometry with confirmation by online HPTLC/electrospray ionization (ESI)–mass spectrometry (MS), ultra-high-performance liquid chromatography (UHPLC)/MS, Fourier-transform infrared (FTIR) spectroscopy, and the proton nuclear magnetic resonance ($^{1}$H-NMR) technique.

## 2. Results

The structure of 5-DCB was proven by UV densitometry (Figure 1), MS analysis (Figures 1–4), FTIR spectroscopy (Figure 5), and the [1]H-NMR technique (Figure 6), and the structural formula was proposed (Table 1).

Figure 1 highlights the HPTLC/UV densitometry analysis of BA, 5-CQA, 5-DCB, the GCB extract, and the GCB extract with BA. In both 254 nm and 365 nm chromatograms, a new band was obtained for the chlorogenoborate sample, and in the GCB extract with added BA, the band corresponding to the 5-CQA turns bright blue in 365 nm UV light. The HPTLC assay uncovered a band at an $R_f$ of 0.78, which corresponded to the fragment ion m/z 715. 5-CQA remained lower at an $R_f$ of 0.45. The 5-CQA standard band appears higher, at approximately $R_f$ of 0.61, due to the lower concentration (Figure 1).

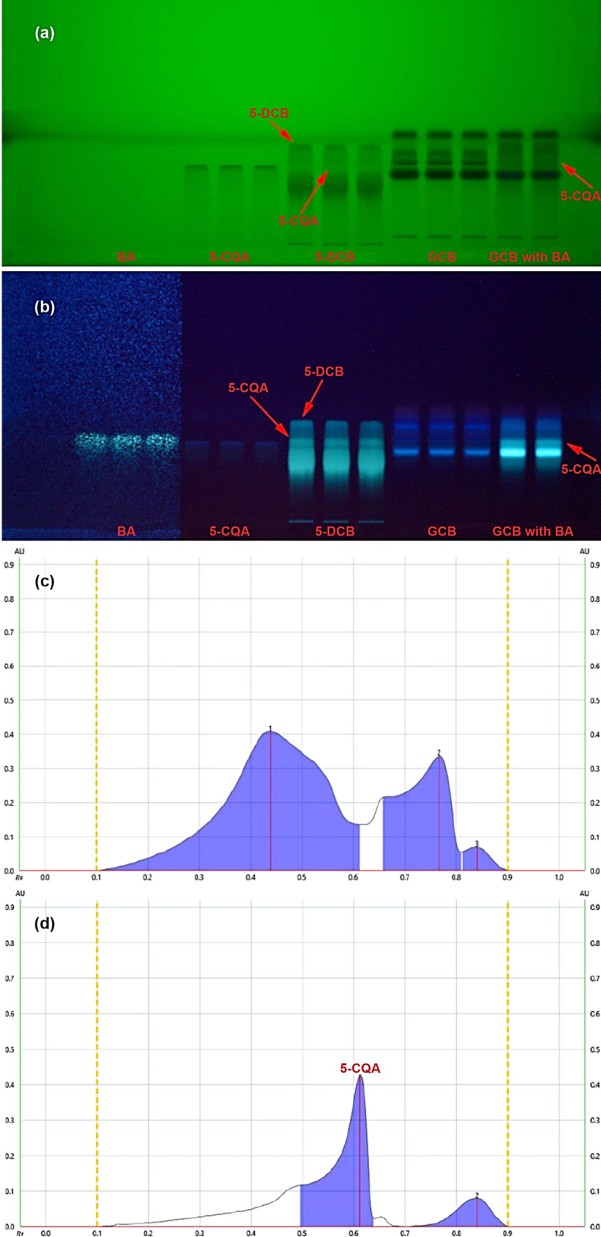

**Figure 1.** HPTLC/UV densitometry: (**a**) HPTLC chromatogram at 254 nm; (**b**) HPTLC chromatogram at 365 nm; (**c**) UV densitogram for 5-DCB complex (peak 2) at 280 nm; (**d**) UV densitogram for 5-CQA at 280 nm. BA: Boric acid; 5-CQA: 5-Caffeoylquinic acid; 5-DCB: Diester chlorogenoborate; GCB: Green coffee bean; HPTLC: High-performance thin-layer chromatography; UV: Ultraviolet.

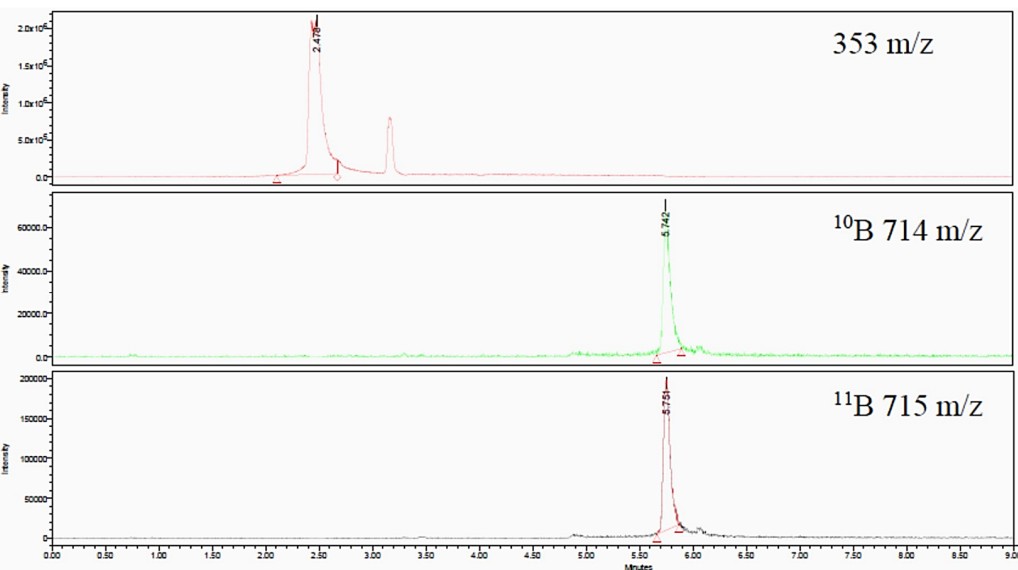

**Figure 2.** UHPLC/MS confirmation of the identified compounds. MS: Mass spectrometry; UHPLC: Ultra-high-performance liquid chromatography.

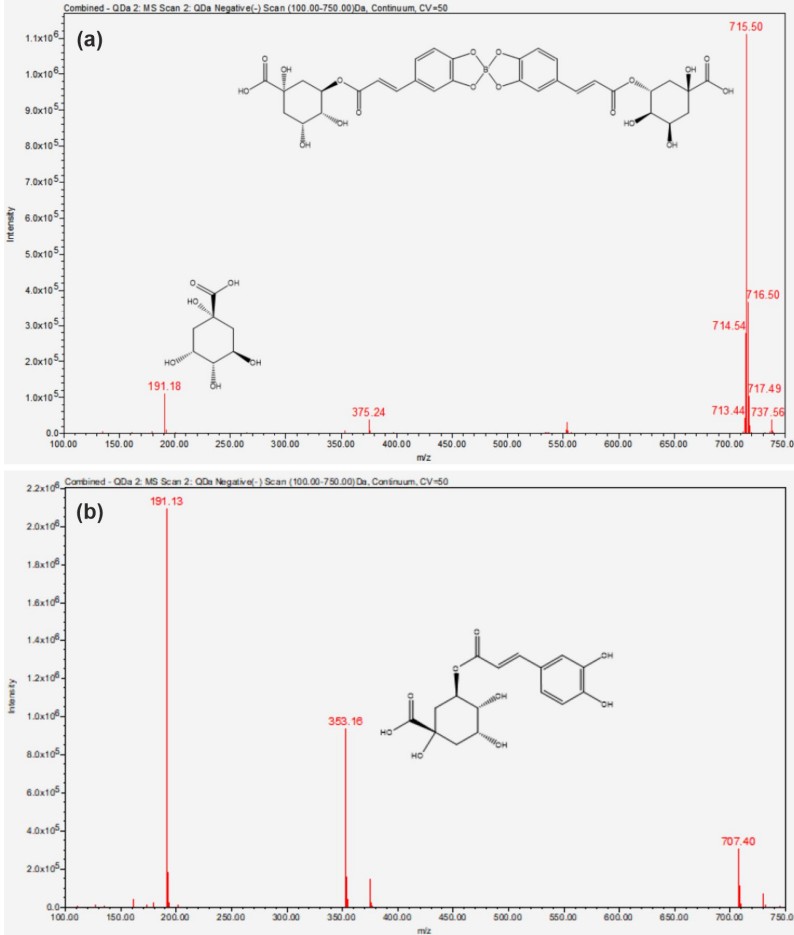

**Figure 3.** Mass spectrometry analysis: (**a**) Mass spectrum of 5-DCB semisynthetic standard (fragment ion m/z 715) with quinic acid fragment ion (m/z 191); (**b**) mass spectrum of 5-CQA (fragment ion m/z 353). 5-CQA: 5-Caffeoylquinic acid; 5-DCB: Diester chlorogenoborate.

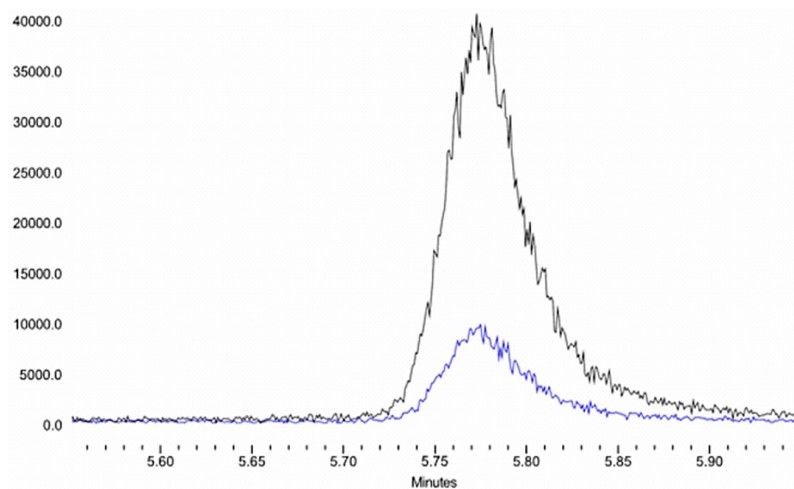

**Figure 4.** 5-DCB complex identified in GCB extract. Blue chromatogram represents [10]B, while the black one highlights [11]B. The ratio between the areas is approximately 1:5, which is corresponding to the specific B isotope ratio. B: Boron; 5-DCB: Diester chlorogenoborate; GCB: Green coffee bean.

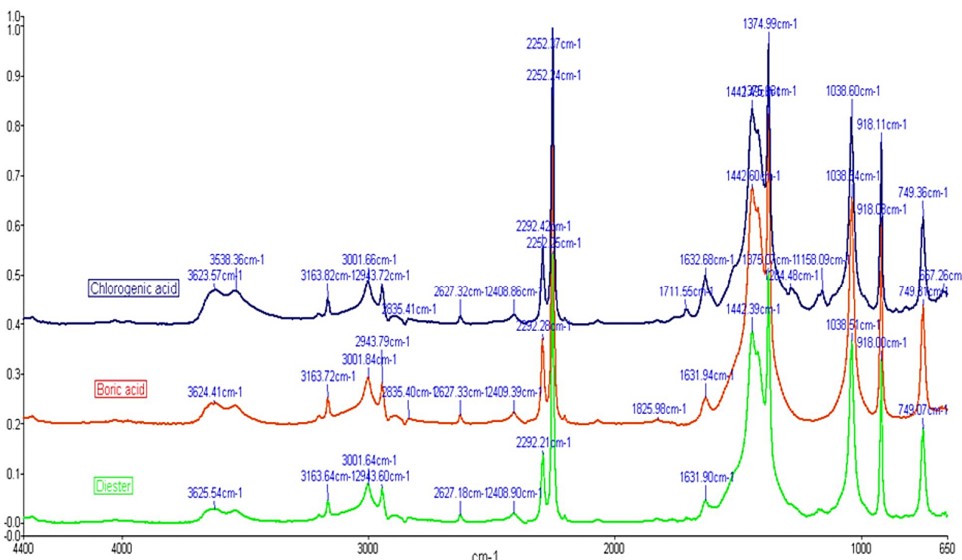

**Figure 5.** FTIR spectrum of 5-DCB complex, BA, and CGA (or 5-CQA). 5-CQA: 5-Caffeoylquinic acid; 5-DCB: Diester chlorogenoborate; BA: Boric acid; CGA: Chlorogenic acid; FTIR: Fourier-transform infrared (spectroscopy).

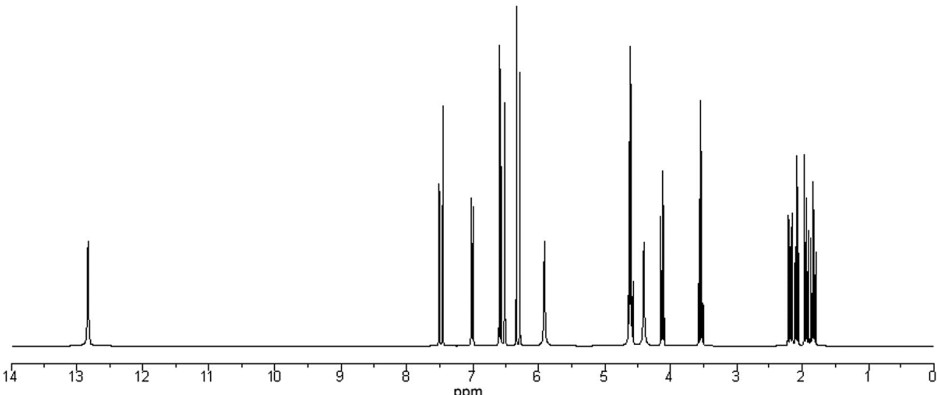

**Figure 6.** [1]H-NMR spectrum of 5-DCB complex. 5-DCB: Diester chlorogenoborate.

**Table 1.** Fragment ions (m/z) found in the mass spectra and supposed molecular structure for 5-DCB complex.

| Compound | Fragment Ions (*m/z*) | Molecular Structure |
|---|---|---|
| 5-CQA | 191 | <br>Quinic acid |
| 5-DCB complex | 353 | <br>5-CQA<br>5-CQA dimer |
| | 707<br>191 | Quinic acid |
| | 715 | <br>Newly formed 5-DCB complex |

5-CQA: 5-Caffeoylquinic acid; 5-DCB: Diester chlorogenoborate.

We identified the 5-DCB peak in selective ion recording (SIR) mode, at 5.75 min, in a concentration of 2.48 µg/g of dry product (Figures 2–4; Table 2). The fragment ion *m/z* 715 also showed *m/z* 714, which was approximately 20% lower than that previously mentioned, indicating a B-containing compound (BCC) (Figure 3a). In Figure 4, the blue chromatogram represents $^{10}$B, while the black one highlights $^{11}$B. The ratio between the areas is approximately 1:5, which corresponds to the specific B isotope ratio.

**Table 2.** RTs and relative areas of the identified compounds.

| Compound | RT [min] | Relative Area |
|---|---|---|
| 5-CQA | 2.478 | 20 921 574 |
| 5-DCB $^{10}$B | 5.742 | 173 776 |
| 5-DCB $^{11}$B | 5.742 | 818 884 |

B: Boron; 5-CQA: 5-Caffeoylquinic acid; 5-DCB: Diester chlorogenoborate; RT: Retention time.

The FTIR spectra of CGA (blue), BA (red), and 5-DCB (green) are vertically scaled in Figure 5, and the following vibrational bands were observed: (*i*) The peaks situated in the range of 3400–3600 cm$^{-1}$ are present for all the compounds, with different intensity ratios (3652/3541) of 1 for BA, 0.95 for 5-DCB, and 0.965 for CGA, with these peaks being assigned to O–H vibration [26–28]; (*ii*) the peaks situated at 2853 and 2922 cm$^{-1}$ are present for all the compounds, being assigned to a C–H torsion vibration [27–29]; (*iii*) the peak situated at

1737 cm$^{-1}$ is characteristic for all compounds, being assigned to an overtone vibration of CH–CH, CH$_2$, CH$_3$, or B$_2$O$_2$ absorption band; the presence of this peak convinced us of the existence of a bond between CGA and B, through the phenyl group [27,28,30]; (*iv*) the peaks situated at 1706 and 1463 cm$^{-1}$ are present only in 5-DCB and CGA spectra, being assigned to an overtone vibration of CH–CH, CH$_2$, and CH$_3$ [30]; (*v*) the peak situated at 1632 cm$^{-1}$ is evidenced for all the compounds, being assigned to the presence of water [27,28,31]; (*vi*) the peak situated at 1520 cm$^{-1}$ is present in the 5-DCB spectrum, but also in the CGA spectrum at 1519 cm$^{-1}$, and is assigned to a vibration rotation of the phenyl bond [32,33]; (*vii*) the peak situated at 1170 cm$^{-1}$ is assigned to an isomeric structure of CGA, being present at 1163 cm$^{-1}$—if the presence of this band is coupled with the presence of 1250 and 1380 cm$^{-1}$, it can be assigned to CGA [29]; (*viii*) the peak situated at 1113.42 cm$^{-1}$ in the 5-DCB spectrum, present in the CGA spectrum at 1120 cm$^{-1}$, is assigned to a trigonelline structure of CGA [30]; (*ix*) the peak situated at 764 cm$^{-1}$ is present only in 5-DCB and BA spectra, being assigned to the existence of a bond between CGA and B, in diester, through the phenyl group [34]; (*x*) some peaks, situated at 817 and 859 cm$^{-1}$, were assigned to the C–O bond from cyclohexane [33]; (*xi*) for the peaks situated at 3650, 859 and 970 cm$^{-1}$ were not found an assignation in the literature.

The newly formed compound exhibits the specific pattern for a B compound: 80% *m/z* 715 [M–H]$^{-}$ for $^{11}$B and 20% *m/z* 714 [M–H]$^{-}$ for $^{10}$B (Figure 3a). For quantification purposes, the SIR for *m/z* 715 was used (Table 1). The limit of detection (LOD) and limit of quantification (LOQ) were 0.396 μg/mL and 1.202 μg/mL, respectively. In Figure 3b, 5-CQA is shown to have a specific fragmentation pattern, with *m/z* 191 [M–H]$^{-}$ for the quinic acid, *m/z* 353 [M–H]$^{-}$ for the 5-CQA, and *m/z* 707 [2M–H]$^{-}$ for the 5-CQA dimer.

The coupling reaction was also discovered to be efficient at room temperature. 5-DCB is readily separated from unreacted 5-CQA by TLC, ion exchange chromatography, or solvent extraction, as is well known in the art. As seen from the chromatograms, a relatively high amount of 5-CQA remains after the reaction and can be identified by the large peak at 2.50 min (Figure 2; Table 2). The newly formed compound can be observed at 5.75 min (Figure 2; Table 2).

The $^1$H-NMR spectrum of 5-DCB is characterized by the chemical shifts $\delta$ (ppm) of protons of the main component: 6.52–7.01 (3H, m, CH aromatic ring) *J* = 0.08; 12.84 (1H, s, COOH); 5.91 (1H, s, OH); 4.4 (1H, s, OH); 4.62 (1H, s, OH); 6.31 (1H, s, CH=); 7.48 (1H, s, =CH); 4.61 (1H, q, CH); 4.13 (1H, t, CH); 3.54 (1H, q, CH); 2.09 and 1.84 (2H, d, CH$_2$); 2.09 and 1.84 (2H, d, CH$_2$) (Figure 6).

Figure 7 highlights the in vitro simulation of 5-DCB digestion: Stomach (gastric phase, at pH 1.2 and pH 4.5) and duodenum (small intestinal phase, at pH 7.2). The digestion of 5-DCB showed a 50% degradation in the stomach acidic medium at pH 1.2, while lower degradation was observed in the intestinal phase at higher pH of 4.5 and 7.2. Subsequently, in a gastric environment of pH 4.5 (after the meal), 5-DCB is protected and works as a classical prebiotic. The same effect occurs when pH 7.2 (similar to the small intestine, the degradation rate is very low) (Figure 7).

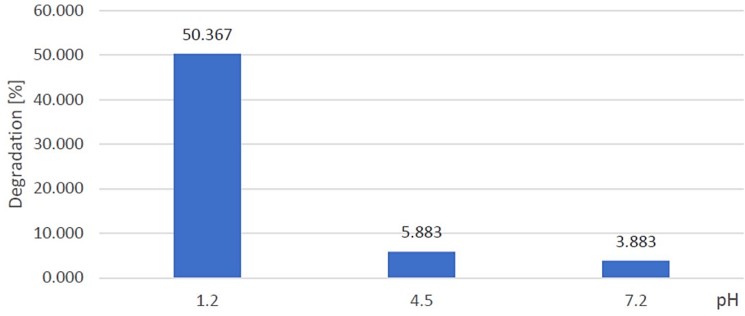

**Figure 7.** In vitro simulation of 5-DCB digestion: Stomach (gastric phase, at pH 1.2 and pH 4.5) and duodenum (small intestinal phase, at pH 7.2). 5-DCB: Diester chlorogenoborate.

## 3. Discussion

B is essential for plants, some bacteria, fungi, and algae; its role has recently been defined as essential in host–bacterial symbiosis in human health [24]. 5-DCB is a new BCC, recently identified in the GCB by our research team. This new compound joins other BCCs, naturally occurring, which have been previously identified, both in plants (pectic polysaccharide borate complex—rhamnogalacturonan II (RG-II), sugar alcohol borate complexes (fructose borate complexes, glucose borate complexes, fructose–sorbitol borate complexes, sorbitol borate complexes, mannitol borate complexes, *bis*-sucrose borate complexes), polyhydroxy organic acid borate esters (malic acid neutral borate complex, mono-malic acid borate complex, *bis* (malic acid) borate complex), amino acid borate esters (*bis*-*N*-acetyl-serine borate complex)) [4], and in bacteria (B complex of vibrioferrin and rhizoferrin or the catechols in petrobactin, the furanosyl borate diester—autoinducer-2 borate (AI-2B), and polyketides—borophycin, boromycin, *N*-acetyl-boromycin, *N*-formyl-boromycin, desvalinoboromycin or TMC 25B, aplasmomycin-A, -B, and -C, monoacetyl-aplasmomycin, and tartrolon B, C, and E) [35].

Most of the known effects of B on human/animal health can be explained by the essentiality of B on healthy symbiosis. The mechanism of action (MoA) of naturally occurring borate species is related both to the B signaling molecule, AI-2B, and to the fortification of the gel layer of the colonic mucus with indigestible B species, prebiotic B natural compounds, such as 5-DCB [24,36,37]. B is an essential element for the growth of certain bacterial species (*Algoriphagus*, *Arthrobacter*, *Bacillus*, *Gracilibacillus*, *Lysinibacillus*, and *Rhodococcus*). The ability of B to attach to glycoproteins is used for symbiosis between bacteria and other kingdoms: Consequently, infection of a symbiotic host organism by the bacteria is blocked. Moreover, B critically influences cell-to-cell signaling during the symbiotic interaction and development of nitrogen-fixing bacteria (*Azorhizobium*, *Bradyrhizobium*, *Rhizobium*) nodules in vegetables. The cell walls of higher plants are stabilized by the B–RG-II polysaccharide complex. In human/animal nutrition, natural organic B compounds could be used as prebiotic candidates. By comparison, because they are digestible and toxic to the microbiota, B inorganic compounds (e.g., BA, borates) cannot be considered prebiotic nutraceuticals. Recent research highlighted the important role of intestinal microbiota and its related metabolites in the protection against the harmful effects of high-dose radiation. In this respect, *Lachnospiraceae* and *Enterococcaceae* bacteria, which need B for cell signaling, have been identified in the gut of mice resistant to radiation [24,36].

Our previous studies demonstrated that, by using HPTLC/UV densitometry with confirmation by online HPTLC/ESI–MS, the identification of nicotinamide riboside and naturally occurring B (NOB) compounds is accurate, precise, rapid, selective, and sensitive [19,38]. For the analysis of the semisynthetic 5-DCB complex and for 5-DCB identification in GCB, the previously developed methods were chosen (HPTLC/UV densitometry with confirmation by online HPTLC/ESI–MS) [19,38] together with UHPLC/MS [24]. The 5-DCB complex was also identified in GCB via the UHPLC/MS technique (Figure 4). Extraction was achieved using 90% acetonitrile, which was then evaporated and resolubilized in the first line of the gradient. The same 1:5 B isotope ratio was observed.

We chose only the plants that had a B concentration higher than 1 ppm and high polyphenols to increase the possibility of identifying new B compounds (especially B-containing polyphenol compounds). After the rigorous tests previously reported, we chose only the GCB that had the highest number of B organic species [25].

The 5-DCB complex, being a B phenolic compound, can be considered a prebiotic, acting as a vector both of B (an essential micronutrient for symbiosis) and 5-CQA [24]. 5-CQA has previously been established as a prebiotic phenolic acid that is only partially digestible in the gastric system [14]. 5-DCB is a promising prebiotic candidate since its pKa is approximately 4.0, and it is indigestible at a pH above the postprandial pH (4.5) of the stomach. In human cells, there are no known biochemical mechanisms that require B and, therefore, B has not found a specific status in the world of nutraceuticals. Subsequently, 5-DCB has become a novel prebiotic candidate and targets the colon as novel colonic food.

Moreover, 5-DCB targets colon nutrition, resulting in a healthy gut microbiome, as well as a healthy microbiome in the mouth, vagina, skin, and scalp [36].

The MoA of the 5-DCB complex is related to both the B signaling molecule (AI-2B) and the fortification of the colonic mucus with B from the specific prebiotic diet. Subsequently, the MoA of B phenolic compounds is the following: The supply of B phenolic compounds allows species of bacteria that require B to communicate using AI-2B, and the fortification of the colonic mucus layer with organic B esters, e.g., borate ester of glycoproteins, helps to protect the host from bacterial infection. Consequently, B phenolic compounds are a source of B, essential for symbiosis, and a source of carbon for the specific nutrition of the microbiota [24]. Given the above-mentioned MoA, both microbiota and the gel layer of the colonic mucus become the 5-DCB target. Therefore, in the future, 5-DCB may become a promising novel prebiotic candidate [36].

Since the specialized literature mentions that the health of the human/animal body depends on the health of the microbiota, the role of B prebiotic compounds becomes essential for the health of the human/animal body in the future [37].

The food-grade 5-DCB complex [36] is potentially useful for (*i*) increasing the buffering capacity of saliva; (*ii*) positively impacting the oral and gut microbiome; (*iii*) protecting significant probiotic bacteria, such as *Lactobacillus* spp. and *Bifidobacterium* spp.; (*iv*) improving SCFAs production; (*v*) improving intestinal barrier integrity and impermeability; (*vi*) improving the immunity system; (*vii*) developing psychobiotic products with nootropic effects; (*viii*) improving antioxidant and anti-inflammatory actions of the microbiome; and (*ix*) providing protection against natural radioactivity in water and soil. Moreover, reagent-grade 5-DCB may be useful for (*i*) a reagent grade ($\geq$95%) adequate for drug, food, or medicinal usage and appropriate for usage in many various analytical applications and (*ii*) some colon-targeted delivery systems for antioxidant therapy in the treatment of inflammatory bowel disease and B neutron capture therapy ($^{10}$BNCT) treatment of colonic cancers with the $^{10}$B-enriched prebiotic 5-DCB complex [36,39]. We were able to synthesize and identify 5-DCB, and the uses of this new natural B complex were patented in the USA [36]. In the future, B phenolic complexes, such as 5-DCB, may be promising new prebiotic candidates [24]. However, the latest investigation proves that in bacteria, the furanosyl borate diester (AI-2B) signaling molecule could contribute to healthy microbiota and its defense against bacteria pathogens [1,2,40]. New perspectives on B's essential role for animals and humans show that a new approach to B is needed to explain B's activity and influence on health [24]. Given this new insight, the essentiality of B species has the potential to provide new possibilities for using the 5-DCB complex in human nutrition to support health and longevity [2,24,41,42].

The new science on B essentiality for a healthy symbiosis between host and microbiome could guide the development of natural B-based dietary supplements to target the microbiota (oral, gut, skin, vaginal, and scalp microbiome) of humans.

The 5-DCB complex is a novel prebiotic candidate, and it does not decompose at pH 4.5 postprandially in the upper gastric system [24]. After completing all the phases of intestinal degradation, we will be able to draw the best conclusions on how to protect the 5-DCB complex in the upper gastric system to reach the colon intact, where it is essential and microbiota accessible. Further, studies show that B organic complexes are digested by microbiota [24,39] and the dissociated phenolics and 5-CQA are further metabolized into functional compounds [16].

The gastric acidic medium usually degrades soluble and insoluble B organic species into B monoesters and diesters. However, many natural B organic monoesters and diesters (polyalcohols, sugar acids, and 5-DCB complex) are resistant to gastric degradation because of their pKa (2.5–5). Likely, in the superior part of the digestive tract, B organic species are reconstituted even at pH 4 (postprandial), starting from the finding that BA is easily associated with *cis*-diol sugars (fructose, ribose, glucose) and phenolic acids [24,36].

The accumulation of B in the intestinal mucus gel layer after the ingestion of the 5-DCB complex can have essential health effects. Exogenous supplementation with a

natural 5-DCB-rich extract increases AI-2B concentrations, reduces intestinal damage, and corrects changes in the microbial flora in diarrhea. Thus, many factors lead to the dysbiosis of the microbiota and significantly decrease the concentration of AI-2B, which leads to the reduction of the ability of the quorum sensing system to maintain the stability of the intestinal microbiome, leading to the potential for the occurrence of diarrhea with major effects on the occurrence of acute and chronic systemic diseases [24,37,43].

## 4. Materials and Methods

### 4.1. Chemicals and Solvents

The 5-CQA standard (98% purity) was acquired from Alfa Aesar (Thermo Fisher GmbH, Kandel, Germany). The BA standard and solvents (2-propanol, acetonitrile, water—LiChrosolv® grade; formic acid—EMSURE® grade) were all acquired from Merck (Burlington, MA, USA). HPTLC Si 60 $F_{254}$ 20 × 20 cm glass plates were also obtained from Merck.

### 4.2. Plant Material and GCB Extract

The product in which we identified and quantified 5-DCB was the Ethiopian Djimmah GCB acquired from an online market. Powdered decaffeinated GCB (10 g) and 100 mL of water were charged in a flask fitted with a stirrer, and the temperature was increased to 55–70 °C, with constant stirring for four hours. Thereafter, 100 mL of acetonitrile was added, and GCB was stirred in 50% aqueous acetonitrile at a temperature of 50–55 °C. The reaction mixture was filtered, and the GCB powder was transferred back to the flask. Extraction with 50% aqueous acetonitrile was repeated 2–3 times until the GCB was completely exhausted. The liquid layer was separated from the GCB residue, and the acetonitrile was evaporated to produce a water solution of GCB extract [36].

### 4.3. Semisynthesis of 5-DCB

To obtain a 5-DCB standard, 5-CQA (Alfa Aesar, Thermo Fisher GmbH, Kandel, Germany) and BA (Merck, Burlington, MA, USA) were used. 5-CQA and BA in a water solution were mixed at 65 °C for three hours. After cooling, the solution was frozen and freeze-dried for 24 h. Then, the solid mass obtained was separated by preparative TLC by utilizing a thick layer of adsorbent (5 mm). Approximately 250 mg of 5-DCB was obtained at a purity of 95% 5-DCB and used to create a calibration curve. The freeze-dried powder was dissolved in water to obtain 5-DCB concentrations of 2 μg/mL, 4 μg/mL, and 8 μg/mL [36].

### 4.4. Methods of Analysis

#### 4.4.1. HPTLC/UV Densitometry with Confirmation by Online HPTLC/ESI–MS

The HPTLC/UV densitometry with evidence from online HPTLC/ESI–MS was performed using a CAMAG (Muttenz, Switzerland) system. The band sample application was accomplished using the CAMAG Linomat 5. Elution was conducted in a 20 × 20 cm twin-trough chamber with a mixture of 2-propanol–water (8:2, $v/v$) with 0.1% formic acid. After the elution, the plate was dried and scanned with the CAMAG TLC Scanner 3 to obtain the densitograms. All the equipment was controlled using the CAMAG VisionCats software package. To confirm the compounds, we used the TLC–MS Interface 2 to directly collect the bands from the plate and introduce them into the mass detector. We tried various mobile phases, and the best yield result was acetonitrile–water (9:1, $v/v$) with 0.1% formic acid.

#### 4.4.2. UHPLC/MS Analysis

The UHPLC/MS analysis was performed on the Waters (Milford, MA, USA) Arc System coupled with a Waters QDa with an ESI probe. The column was a Waters Cortecs C18 (4.6 × 50 mm, 2.7 μm) eluting with two solvents: A (0.1% formic acid in water) and B (0.1% formic acid in acetonitrile). The gradient used was as follows: 0–10 min 10% to 50% B, 10–11 min 50% to 10% B. The mobile phase had a flow rate set at 0.6 mL/min. The column temperature was equilibrated to 20 °C. The injection volume was 5 μL. The QDa

was set to negative mode at 0.8 kV for the capillary, 50 V for the cone voltage, and 400 °C for the capillary. The mass range was set at $m/z$ 100–800. The photodiode array (PDA) was set to only detect the 325 nm wavelength. A relatively high amount of 5-CQA remained after the reaction and could be identified by the large peak at 2.50 min.

### 4.4.3. FTIR Spectroscopy

A Frontier Two FTIR spectrophotometer, monitored by a distinctive software, Spectrum v. 10.5.1 (Perkin Elmer, Inc., Boston, MA, USA), was utilized for analysis. The software permits both quantitative (Spectrum Quant) and qualitative (Spectrum Search) analysis, by detecting the specific bands in comparison with equivalent complexes in the database. The quantitative analysis used the Lambert–Beer law at a certain wavelength or chemometric methods (Principal Component Regression (PCR), Partial Least Squares (PLS), or PLS+) for determination, in which the entire obtained spectral range can be used. The samples were examined using the Attenuated Total Reflection (ATR) method, with a ZnSe crystal, which permits recording the IR spectrum in the range of 4000–650 cm$^{-1}$, or the KRS-5 method, which permits recording the IR spectrum in the range of 4000–400 cm$^{-1}$. In the above-mentioned IR ranges, 16 scans were performed, with a resolution of 4 cm$^{-1}$.

### 4.4.4. $^1$H-NMR

$^1$H-NMR spectra were achieved on a VnmrS 400 NMR spectrometer (Varian, Palo Alto, CA, USA) at 400 MHz (DMSO-$d_6$). The chemical shifts were recorded against an internal standard (tetramethylsilane).

### *4.5. In Vitro Simulation of 5-DCB Digestion*

In vitro digestion of 5-DCB was performed using a human gastric simulator for the stomach (gastric phase) and duodenum (small intestinal phase), according to our previously published paper [24].

### 5. Conclusions

In our opinion, 5-DCB is of great importance for human health, as it has been shown to be an effective prebiotic essential in the healthy symbiosis between the microbiome and the human host. Furthermore, due to its B content, 5-DCB is likely a potential prebiotic essential for human nutrition. For the first time, we report that a new natural B organic complex, 5-DCB, was obtained by semisynthesis and then identified in GCB. 5-DCB may be a promising new prebiotic candidate to target the human/animal microbiome. New knowledge about the essentiality of B species for a healthy symbiosis between human/animal hosts and microbiota will lead to the use of natural B-based nutraceuticals, such as 5-DCB, to target the human/animal microbiome. Of these, the gut microbiome is the most important for human health. Subsequently, B prebiotic species have become novel prebiotic candidates and target the colon as novel colonic foods. Moreover, B species target colon nutrition, resulting in a healthy gut microbiome, as well as a healthy microbiome in the mouth, vagina, skin, and scalp. Consequently, 5-DCB, as a B prebiotic compound claimed to be nutritionally essential for the symbiosis between the microbiota and the host, plays a role in the prevention of certain diseases, such as osteoarthritis, osteoporosis, rheumatoid arthritis, cardiovascular diseases, thyroid diseases, depression, oral diseases, obesity, diabetes, viral, bacterial, and parasitic infections. This paper only considers the identification of 5-DCB, and not isolation and characterization of its molecular structure. In the future, we intend to isolate 5-DCB from plants and to completely characterize it from the physicochemical point of view.

**Author Contributions:** Conceptualization, A.B. (Andrei Biţă) and I.R.S.; formal analysis, N.R., L.E.B. and G.R.; investigation, C.B., M.V.C., A.B. (Andrei Bunaciu) and M.I.P.; resources, A.B. (Andrei Biţă), I.R.S. and J.N.; writing—original draft preparation, A.B. (Andrei Biţă), I.R.S. and G.D.M.; writing—review and editing, I.R.S., L.D. and G.D.M.; visualization, I.R.S., J.N. and G.D.M.; supervision, I.R.S., J.N. and O.C.R.; funding acquisition, I.R.S. and J.N. All authors have read and agreed to the published version of the manuscript.

**Funding:** This work was supported by a grant from the Ministry of Research, Innovation and Digitization, CCCDI–UEFISCDI, project number PN-III-P2-2.1-PED-2021-0804, within PNCDI III.

**Institutional Review Board Statement:** The manuscript does not contain experiments on laboratory animals.

**Informed Consent Statement:** The manuscript does not contain clinical studies or patient data.

**Data Availability Statement:** Publicly available datasets were analyzed in this study. This data can be found here: [https://drive.google.com/drive/folders/1ekB61adWPzX1iYS6UE6bEkcuAMoL2C7Z?usp=share_link] accessed on 4 March 2023.

**Acknowledgments:** We thank Jennifer Murphy, Director of Innovation and Clinical Development from PLT Health Solutions, Inc., for her kind assistance with the final version of the manuscript.

**Conflicts of Interest:** The authors declare no conflict of interest. The funders had no role in the design of the study; in the collection, analyses, or interpretation of data; in the writing of the manuscript; or in the decision to publish the results.

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
