# Peer review of "Diester Chlorogenoborate Complex: A New Naturally Occurring Boron-Containing Compound"

_inorganics, doi:10.3390/inorganics11030112_

Round 1
Reviewer 1 Report
Conclusion summarized the results and discussions in a concise, effective manner, but need be more clear. For example, "obtained by semisynthesis and then discovered in plants (GCB)" is not accurate here.
Author Response
Dear Reviewer,
First of all, we would like to address You many thanks for your accurate observations and valuable comments. We used all these and improved the paper accordingly. All changes in the revised manuscript were marked up using the “Track Changes” function. The following changes have been made for the Manuscript (ID inorganics-2229142):
Reviewer #1 question/comment:
Conclusion summarized the results and discussions in a concise, effective manner, but need be more clear, e.g., “obtained by semisynthesis and then discovered in plants (GCB)” is not accurate here.
Answer:
The sentence has been rephrased accordingly: “… was obtained by semisynthesis and then identified in GCB… New knowledge about the essentiality of B species for a healthy symbiosis between human/animal hosts and microbiota will lead to the use of natural B-based nutraceuticals, such as 5-DCB, to target the human/animal microbiome. Of these, the gut microbiome is the most important for human health. Subsequently, B prebiotic species have become novel prebiotic candidates and target the colon as novel colonic foods. Moreover, B species target colon nutrition, resulting in a healthy gut microbiome, as well as a healthy microbiome in the mouth, vagina, skin, and scalp.” (See lines 458–466).
Reviewer 2 Report
1. I cannot check the data clearly, thus the quality of all the figures could be boosted.
2. “Figure 5. FTIR spectrum” why the authors give more sentences on the figure caption, the authors should discuss it one separated part. Some the work on discussion of the functional groups could be cited, such as: Inorganics, 10(2022) 202 and Micropor. Mesopor. Mat, 341(2022) 112098.
3. Also, the functional groups from the main peaks should be marked in Fig5.
4. “5-DCB complex is a novel prebiotic candidate, and it does not decompose at pH 4.5..” why it is a novel complex?
5. I suggest the authors test more data for checking the current 5-DCB complex. such as UV, ICP and EA?
6. The authors could describe each part and then put the related figure at the end of the part, there are many figures in one space.
7. The typesetting work should be checked in the revision.
Author Response
Dear Reviewer,
First of all, we would like to address You many thanks for your accurate observations and valuable comments. We used all these and improved the paper accordingly. All changes in the revised manuscript were marked up using the “Track Changes” function. The following changes have been made for the Manuscript (ID inorganics-2229142):
Reviewer #2 questions/comments:
1. I cannot check the data clearly, thus the quality of all the figures could be boosted.
Answer:
The quality of the Figures (1–7) has been boosted during the manuscript revision. All images were processed as *.jpg files with a 300-dpi resolution.
2. “Figure 5. FTIR spectrum” why the authors give more sentences on the figure caption, the authors should discuss it one separated part. Some the work on discussion of the functional groups could be cited, such as: Inorganics, 10(2022) 202 and Micropor. Mesopor. Mat, 341(2022) 112098.
Answer:
Interpretation of vibrational bands from FTIR spectra has been discussed in one separate part. Also, nine citations ([26] to [34]) have been added on discussion of the functional groups, including [27] (Qin et al., Inorganics, 2022) and [28] (Qin et al., Microporous Mesoporous Mater., 2022). (See lines 133–170 & 550–572).
3. Also, the functional groups from the main peaks should be marked in Fig. 5.
Answer:
The functional groups from the main peaks have been marked in the revised Figure 5.
4. “5-DCB complex is a novel prebiotic candidate, and it does not decompose at pH 4.5…” why it is a novel complex?
Answer:
5-DCB is a B prebiotic complex, which has been identified for the first time by our research group.
5. I suggest the authors test more data for checking the current 5-DCB complex, such as UV, ICP and EA?
Answer:
Thank you for your suggestion. The aim of our research was the identification of a new natural organic boron compound. In our future paper, we will approach all aspects regarding its physico-chemical and biological properties.
6. The authors could describe each part and then put the related figure at the end of the part, there are many figures in one space.
Answer:
The related Figures (1–7) have been put immediately at the end, after the description of each part.
7. The typesetting work should be checked in the revision.
Answer:
The typesetting work has been checked during the manuscript revision.
Kind regards,
Ion Romulus SCOREI, Professor, MD, PhD
Corresponding Author
Reviewer 3 Report
After reading and evaluating the submitted article, I would like to state that the article is written clearly and is of interest to the scientific community. However, the NMR spectrum of 5-DCB is not shown.
Remarks:
1. line 209: ...atomic protons...?,
2. Use lines 150-154 instead of the lines 208-213 which are excessive
3. The authors used D2O as a solvent. How they could find signal of OH proton (Line 152) and COOH (line 210)?
4. NMR spectrum must be shown in full.
Conclusion: manuscript requires a minor revision.
Author Response
Dear Reviewer,
First of all, we would like to address You many thanks for your accurate observations and valuable comments. We used all these and improved the paper accordingly. All changes in the revised manuscript were marked up using the “Track Changes” function. The following changes have been made for the Manuscript (ID inorganics-2229142):
Reviewer #3 questions/comments:
After reading and evaluating the submitted article, I would like to state that the article is written clearly and is of interest to the scientific community. However, the NMR spectrum of 5-DCB is not shown. Conclusion: manuscript requires a minor revision.
Remarks:
1. Line 209: ...atomic protons...?
Answer:
Line 209 of the original manuscript has been removed.
2. Use lines 150–154 instead of the lines 208–213 which are excessive
Answer:
Lines 208–213 of the original manuscript have been removed.
3. The authors used D2O as a solvent. How they could find signal of OH proton (Line 152) and COOH (line 210)?
Answer:
The solvent used was DMSO-d6 (and not D2O). The manuscript has been revised accordingly. (See line 449).
4. NMR spectrum must be shown in full.
Answer:
The 1H-NMR spectrum of 5-DCB complex has been showed in full (the newly added Figure 6). (See lines 210–212).
Kind regards,
Ion Romulus SCOREI, Professor, MD, PhD
Corresponding Author
Reviewer 4 Report
Аrticle " Synthesis and Innovative Biological Activity of Boron-Containing Compounds" Authors: Andrei Biţă , et all. The article discusses the analysis of the structure of the natural diester chlorogenoborate complex, as well as its possible applications in hygiene products and the food industry. The article is structured, written in a clear and understandable language, the literature corresponds to the stated topic. However, sections: discussion of results, conclusions do not reflect the stated goals of the work.
There are two main research messages in the article.
The first message: there are natural boron-containing compounds that are useful for humans. These compounds are found in plants and must be isolated and identified. In this part of the work - all is well.
The second message is that certain isolated natural boron-containing compounds can be synthesized. In this part of the work, data from studies of the synthesis of such compounds should have been given - such information is not available..
Also, the authors write a lot about the biological properties of the boron-containing compounds they study. Such information is useful in the "introduction" section. Because the article does not contain detailed studies on the effect of boron-containing compounds on biological activity, safety for a living organism, then information on the biological properties of boron-containing compounds is redundant for the "discussion" section
The article lacks information about the synthesis: conditions, concentrations, ratios of reagents, degrees of conversion, quantitative yield of the product, etc.
The conclusions are general. They do not reflect the results of both the isolation of boron-containing compounds and their synthesis.
I would recommend publishing the article after it has been finalized.
Author Response
Dear Reviewer,
First of all, we would like to address You many thanks for your accurate observations and valuable comments.
All details regarding the Reviewer requirements are found in USPTO Patent Application No. PCT/US22/78488 (from 10/21/2022), which is under review by United States Patent and Trademark Office (USPTO); therefore, we cannot disclose details regarding the synthesis of the new natural organic B complex. Moreover, our article aims to communicate only the identification of this new natural B compound.
Regarding the isolation of the new natural B compound, our paper only considers the identification of this compound. Issues regarding the isolation of the new B compound are an integral part of the patent under evaluation. Please see the Reference No. [36]:
- Scorei, I.R.; Bita, A.; Dinca, L.; Mogosanu, G.D.; Rangavajla, N. Borate complexes of chlorogenic acid and uses thereof. United States Patent and Trademark Office (USPTO), Patent Application No. PCT/US22/78488 from 10/21/2022. [https://www.uspto.gov/patents]
Kind regards,
Ion Romulus SCOREI, Professor, MD, PhD
Corresponding Author
Round 2
Reviewer 2 Report
work on the comment clearly
Author Response
Dear Reviewer,
Once again, we thank You for your review, which helped us a lot to improve the quality of our manuscript.
Kind regards,
Ion Romulus SCOREI, Professor, MD, PhD
Corresponding Author
Reviewer 4 Report
Dear authors!
I would like to draw your attention to the fact that the stated purpose of the study :
"We present here a method for 5-DCB semisynthesis and rapid identification from 93 GCB by using the high-performance thin-layer chromatography (HPTLC)/ultraviolet 94 (UV) densitometry with confirmation by online HPTLC/electrospray ionization (ESI)–95 mass spectrometry (MS), ultra-high-performance liquid chromatography (UHPLC)/MS 96 and proton nuclear magnetic resonance (1H-NMR) technique."
In accordance with the purpose of the study, the "Discussion" section should provide information on the features of extraction and identification of the boron-containing object. This information should clearly reflect the novelty of the research and should be the main one in this section.
Section "Discussion". At present аuthors write a lot about the biological properties of the boron-containing compounds they study. Such information is already in the "Introduction" section. The "Discussion" section looks like a short review article about the biological properties of the boron-containing complex. This is at odds with the purpose of the study. I would recommend shortening the biological part of this section and adding more discussion on the "isolation" and analysis of the boron compound. In addition, in the "Discussion" section, the authors did not discuss Figure 3b, Figure 4-7.
"Conclusions" must be presented in accordance with section "Discussion".
According to patent law, once a patent application has been filed, authors have no restrictions on publication.
Author Response
Dear Reviewer,
First of all, we would like to address You many thanks for your accurate observations and valuable comments. We used all these and improved the paper accordingly. All changes in the revised manuscript were marked up using the “Track Changes” function. The following changes have been made for the Manuscript (ID inorganics-2229142):
The “Discussion” section has been rephrased to clearly reflect the novelty of the research. (See lines 273–291, 336–355, 536–543).
In the “Discussion” section, the biological part has been shortened accordingly, eight paragraphs being removed from previously revised [Round I] manuscript.
Figures 3–7 have been described accordingly in the “Results” section. (See lines 123–127, 197–201, 215–219).
The information on the features of extraction and identification of the new natural B organic complex has been added in the “Materials and Methods” section. (See lines 552–559, 561–568, 684–686).
The “Conclusions” were duly completed and presented in accordance with the “Discussion” section. (See lines 708–711, 720–724).
We have also introduced other additions/modifications that we hope will improve the quality of the manuscript:
▪ Other corrections have been added to improve the quality of the manuscript. (See lines 52, 53, 55, 64, 96, 97, 101–103, 105–110, 157, 158, 169).
▪ Two new References relevant to the contents of the manuscript have been added: Ref. [35] (Dembitsky & Gloriozova, 2017) and Ref. [37] (Mitruţ et al., 2022). (See lines 830, 831, 835–837).
▪ Ref. [35] and Refs. [37] to [41] of the previously revised [Round I] manuscript have been renumbered accordingly: Refs. [38] to [43]. (See lines 838–853).
Kind regards,
Ion Romulus SCOREI, Professor, MD, PhD
Corresponding Author